# The Relationship between Social Frailty and Depressive Symptoms in the Elderly: A Scoping Review

**DOI:** 10.3390/ijerph192416683

**Published:** 2022-12-12

**Authors:** Xiaojing Qi, Jie Li

**Affiliations:** School of Nursing, Tongji Medical College, Huazhong University of Science and Technology, Wuhan 430074, China

**Keywords:** social frailty, depressive symptoms, the elderly, scoping review

## Abstract

Background: Various studies have highlighted the correlation between social frailty and depressive symptoms in the elderly. However, evidence of how these two domains influence each other is not clear. The purpose of this scoping review is to summarize the current literature examining social frailty and depressive symptoms. Method: We conducted a scoping review allowing for the inclusion of multiple methodologies to examine the extent and range of this research topic. Result: The search initially yielded 617 results, 14 of which met the inclusion criteria. Five studies were identified from China, six were identified from Japan, two were identified from Korea, one was identified from Ghana, and one was from Asia. The evidence reviewed indicated that five studies met category 5 criteria, and the others met level 3 criteria. The findings from these studies showed that there is a significant relationship between social frailty and depressive symptoms. Conclusion: This scoping review shows that worse social frailty contributes to a significant degree of depression. Further research on screening social frailty and possible interventions in community and medical settings to prevent the elderly from developing depressive symptoms is needed.

## 1. Introduction

Depressive symptoms and social frailty are increasingly popular in geriatric research. The relationship between the two topics is of great interest to researchers. Depressive symptoms are common among elderly adults, and are associated with adverse health outcomes, such as functional disability and mortality [1,2]. Depressive disorders are expected to be the second most common cause of disability worldwide by 2030, and the leading cause in high-income countries [3]. It is thus critical to prevent depressive symptoms and associated factors that are amenable to intervention.

One such factor is social frailty. Social frailty is one of the social domains of frailty, which is a common geriatric syndrome. The concept of frailty is widely regarded as a multidimensional construct with physical, cognitive, psychological, and social components [4]. However, most previous studies have emphasized assessments of the physical domain of frailty, excluding the social domain. Social frailty is a notable domain concerning the elderly. This is based on evidence that is associated with clinical outcomes, such as functional disability and the risk of mortality among the elderly [5,6,7]. Recently, a systematic review showed that not only physical and cognitive aspects, but also social aspects of frailty are essential components of health in the elderly. In fact, social frailty has been found to lead to various health problems later in life [8]. This issue has been highlighted by the COVID-19 pandemic, which has had a significant negative effect on the mental health of older adults, even though this segment of the population seems to have been more resilient than younger groups [9].

A previous study reported that social frailty was associated with new-onset depressive symptoms after a 4-year follow-up [8]. In a longitudinal study, after adjusting for sociodemographic factors (e.g., age, gender, smoking, alcohol intake, etc.), social frailty and pre-social frailty were associated with a 2.31-fold and 1.58-fold increased risk of incidence of depressive symptoms, respectively, compared with non-socially frail participants [10]. In a cross-sectional analysis of one prospective cohort study, social frailty was associated with depression; that is, participants with depression had a higher prevalence of social frailty [11]. Thus, studies have consistently found self-reported poor social frailty among elderly members with depressive symptoms. 

Past research has examined social frailty as a predictor of, or correlated to, depressive symptoms among older adults. However, findings regarding the potential relationship are ambiguous, since investigators have used multiple approaches to measure social frailty and depressive symptoms which has produced inconsistent variables and results. Prior reviews can be used to guide researchers to develop new or refined intervention strategies through empirical research [12]. In this context, this review can be used as a reference for researchers who wish to seek preventive measures for depressive symptoms. This topic is particularly important given that the current strategies for old-age income security in all types of countries, based on fostering private pensions, do not seem particularly successful thus far in terms of encouraging household savings that warrant decent ageing [13,14].

## 2. Methods

The scoping review method is an approach that allows for the inclusion of diverse methodologies (i.e., experimental and non-experimental research) and has a great impact on evidence-based practice in nursing. The approach can be used to map fields of a topic where it is difficult to visualize the range of material category, which contributes to the presentation of varied perspectives on a phenomenon of concern. We followed the steps proposed by Arksey and O’Malley [15] to conduct this review, which includes 5 stages: identifying the research question; identifying relevant studies; study selection; charting the data; and collating, summarizing and reporting the results.

A thorough and exhaustive review of the literature was completed, and there is no review addressing our research question. We did not intend to complete further meta-analysis or sub-group analysis due to the heterogenicity of the study designs included in this review. The leveling of evidence is based on Melynk and Fineout-Overholt’s rating system for hierarchies of evidence [16]. 

### 2.1. Identifying the Research Question

The purpose of this scoping review of the literature is to examine the literature to integrate and compare findings and further explore the relationship between social frailty and depressive symptoms.

### 2.2. Identifying Relevant Studies

The literature search was conducted using PubMed, Web of Science, CINAHL, EMBASE, Ovid MEDLINE, and ProQuest (Table 1). The reference lists of the identified studies were also hand-searched for any other relevant studies.

### 2.3. Search Strategy and Study Selection

An interchangeable combination of the following keywords was used to search for pertinent literature. The search terms were social frailty and depres*. The search was limited to primary research studies published from the establishment of the database up to June 2022. Studies reporting the relationship between social frailty and depressive symptoms were included in this review. Moreover, only those published in English and Chinese were selected due to the language limitations of the authors and cost limitations.

### 2.4. Charting the Data

Data were extracted into a piloted Excel document. The main items that were extracted from each study included: first author, year of publication, purpose, study design, sample, setting, variables, measure tools, cutoffs of scales, control variables, major findings, and follow-up time (if cohort study). Extracted data were compared item by item so that similar data were categorized and grouped, then coded categories were compared to further analysis and synthesis process. The data from these matrices are summarized in Appendix A Table A1 and Table A2.

### 2.5. Collating, Summarizing and Reporting the Results

The included studies were organized into 3 categories: cross-sectional studies, prospective cohort studies, and retrospective cohort studies. For each of these 3 categories, the primary finding in this study was the correlation between social frailty and depressive symptoms. Studies that were selected using this retrieval procedure were assessed for methodological quality using standardized critical appraisal instruments from the Joanna Briggs Institute (JBI) [17].

## 3. Results

Based on database searches, 14 studies were found. The main reason for inclusion was studies reporting on the target primary outcome (that is the relationship between social frailty and depression). All kinds of designs are included in this integrative review.

### 3.1. Characteristics of Studies Included

Table A1 and Table A2 show the characteristics of the studies included in the review. Of the 14 original studies, 9 are cross-sectional studies [4,11,18,19,20,21,22,23,24], 3 are prospective cohort studies [8,25,26], and 1 is a retrospective cohort study [27]. One analyzed both cross-sectional data and data from prospective cohort studies [10]. Five studies were conducted in China, six in Japan, two in Korea, one in Ghana, and one in Asia. Three studies were population-based, ten were community-based, and one was conducted at a single university hospital. None of the cohort studies were non-randomized controlled prospective studies or non-randomized controlled retrospective studies; thus, according to the Canadian Medical Association and Center for Evidence-Based Medicine (2001), evidence from 5/14 studies can be categorized as level 5, while the other studies can be categorized as level 3.

### 3.2. Defining, Conceptualizing and Measuring Social Frailty

Eight studies defined social frailty. These studies incorporated and explicated the concept of social frailty based on Bunt’s [28] conceptual model of social frailty in a scoping review. They defined social frailty as the absence of crucial general and social resources, social behaviors and activities, as well as self-management abilities vital for achieving one’s social needs and invariably negatively impact subjective well-being during the lifespan. However, another four studies did not provide a definition for social frailty, though they were of appropriate rigor in spite of this omission. The provision of conceptual definitions provides clarity and trustworthiness to studies. In contrast, the absence of conceptual definitions and clarity perpetuated confusion regarding measurements and conclusions [29].

Screening for social frailty is not widely implemented, largely owing to controversial measurements. Of the fourteen studies, the measure tools for social frailty were quite different ranging from two to nine items, which will result in different cutoffs and outcomes. All of them were very brief, except for one with nine items. Ten of the fourteen studies simply used the “yes/no” answer. Furthermore, researchers infrequently explored the validity and reliability of their own scales. Only Pek, K [4] delineated the construct validity of the social frailty brief scales through empirical statistical techniques, i.e., exploratory factor analysis (EFA), to derive a social frailty scale grounded in Bunt’s proposed conceptual framework. He interpreted the retained three factors of ‘social resources’, ‘social activities and financial resource’, and ‘social need fulfillment’, which addressed various components when mapped onto Bunt’s social frailty concept. Similarly, this article does not verify the reliability of the scale.

Given the limitations of the different and brief measures, binary (yes or no) scales, and lack of reported reliability and validity, the strength and accuracy in measuring the construct of social frailty is questionable. However, this does not influence the strong predictive function of adverse health outcomes, indicating that the scale is a useful tool [11].

### 3.3. Operationalizing and Measuring Depression

A variety of instruments were used to measure depressive symptoms. Most studies applied the Geriatric Depression Scale (GDS-15), and one used the brief measures Geriatric Depression Scale (GDS-5). Although the item number of the GDS-5 is more concise than the GDS-15, it is as effective as the 15-item GDS for depression screening in community-dwelling populations of 65 years or older, and may prove to be a preferred screening test for depression [30]. Moreover, 4/11 studies used the Center for Epidemiologic Studies Depression Scale (CESD) to assess depressive symptoms.

Geriatric Depression Scale (GDS-15): The GDS-15 is an instrument that elicits self-reported responses about depressive symptoms, and contains 15 questions, ranging from 0 to 15. The GDS is unique in that it has been specifically developed for geriatric patients. There is no study checking its reliability or validity since it is one of the most widely used scales for depression detection. The GDS suggests an optimal cutoff score of 4/5 (≥5 meaning optimally identified depression) [31]. However, of the nine studies applying the GDS-15, researchers judged the cutoff at different points, even in the same country. For example, Maho Okumura classified scores ≥5 on the GDS-15 as “new-onset of depression”, while Kota Tsutsumimoto PhD regarded this category as ≥6 for Japanese older adults [8,27]. Additionally, ZeKun Chen equated scores ≥6 on the GDS-15 with showing depressive symptoms, but Wenya Zhang considered this marker as scores ≥8 in Chinese older adults [10,22]. While the cut offs were slightly different between studies, they were largely very similar.

Center for Epidemiologic Studies Depression Scale (CESD): Each item was rated on a 4-point Likert scale, ranging from “rarely or none of the time (0–1 day)” to “most or all of the time (5–7 days).” Only one study employing the CESD reported validity measurements with a principal component analysis, which yielded two factors, including negative affect and positive affect. Meanwhile, this study showed that the internal consistency of the CESD-10 was good, with a Cronbach alpha of 0.79 [25]. However, validity and reliability were not specified in the rest of the studies using the CESD.

Other measures. One of the eleven studies regarded depressive symptoms as both self-reported cases and self-reported symptoms after being diagnosed with depression or meeting the International Classification of Diseases, 10th revision (ICD-10) criteria for a mild depressive episode based on the reported symptoms. The ICD-10 is a validated and widely used medical classification of diseases and health-related problems offered by the World Health Organization [18]. Medical diagnosis is reliable as well, but self-reporting can produce some errors due to the sense of shame that some participants experience. Another Japanese study applied the 6-item Kessler scale (K6) to assess the symptoms experienced in the last 30 days and regarded scores ≥5 as showing depressive symptoms. This has been shown to have excellent sensitivity [24].

### 3.4. Identify the Relationship between Social Frailty and Depressive Symptoms

#### 3.4.1. Cohort Studies

Five cohort studies revealed all significant relationships between social frailty and depressive symptoms by χ^2^, logistic regression analyses and Cox regression models. Of these, three were prospective studies and one was a retrospective cohort study. The dependent variable was depressive symptoms. Notably, the shortest follow-up period was only 1 year [27], which is likely to limit the ability to identify temporal changes in the incidence rate of depressive symptoms. However, there were positive relationships between social frailty and depressive symptoms in all of the findings, even though the shortest follow-up time was only 1 year.

Although these studies provide support for the relationship between social frailty and depressive symptoms, some findings are more trustworthy than others. For example, 4/5 cohort studies were from large populations or community-based longitudinal populations with a sample size of 1103–6641, such as the aging arm of the Rugao Longevity and Aging Study (RuLAS), the National Center for Geriatrics and Gerontology (NCGG) Study of the Geriatric Syndrome database, and the China Health and Retirement Longitudinal Study (CHARLS). These large samples resulted in small, expected sampling errors [32]. Furthermore, two of them chose more advanced sampling methods. For example, Lian, Y and colleagues [25] used a multistage sampling strategy for 28 provinces, 150 county-level units, and 450 communities/villages. Chen, Z and colleagues [10] chose participants according to 5-year age and gender strata, which can improve the accuracy of the sample relative to the population [33]. Additionally, gender strata sampling is a probability sampling method, of which the random selection of subjects is best known as the accepted way of achieving representativeness. In probability samples, each member of the population of interest has a known non-zero chance of inclusion [34]. The use of a large longitudinal population and a reliable sampling method are strengths. However, on the other hand, Okumura, M and colleagues [27] chose only 48 subjects to investigate the impact of preoperative social frailty in older patients with gastrointestinal cancer on new-onset depressive symptoms 1-year post-surgery. The limitations of this study were the small sample size and selection bias, but the authors accounted for the shortage by adjusting for the appropriate confounding factors and comparing both those patients who were analyzed at follow-up and those who were not.

All the subjects were chosen at a similar point. Any causes of attrition were described in each individual study, and 3/5 cohort studies provided strategies to address incomplete follow-up by comparing the patients who were analyzed at follow-up and those who were not, as described above, or impute missing values by applying the approach of multiple imputation to impute missing data at the follow-up assessment [8,24,27].

All of the five cohort studies adjusted for control variables, confounding the predictive function for depressive symptoms of social frailty when performing statistical analysis. These control variables included age, sex, body mass index, education, medication, medical history, physical activity, GDS score at baseline, etc.

#### 3.4.2. Cross-Sectional Studies

Nine cross-sectional studies showed an existing relationship between social frailty and depressive symptoms (see Appendix A Table A2). As for the statistical analysis methods, 2/9 applied χ^2^ to compare whether there was a significant difference in the prevalence of depression in each social frailty group, while 5/9 used logistic regression, indicating that social frailty independently increased the risk of depressive symptoms. One used the R caret package to assess the importance of social frailty to predict depressive symptoms via AUC, indicating that these predictors can discriminate more effectively than a coin toss [20].

Other studies lacked appropriate control variables and used less sophisticated statistical analyses. For example, 4/9 studies did not identify or cope with confounding factors or include any control variables. However, control variables may affect the correlation between social frailty and depressive symptoms. For example, one study discovered a pathway of “socioeconomic status—social frailty—depressive symptoms”, showing that lower socioeconomic status may result in vulnerability to the detrimental effect of social frailty on depressive symptoms [25].

Two studies drew conclusions based solely on correlation analysis, demonstrating that depressive symptoms are significantly correlated to social frailty, and that the five items of the social frailty scale (i.e., lack of social role, insufficient social participation, loneliness, economic difficulties, and reduced conversation with others) are positively correlated with depressive symptoms [21,22].

All of the nine cross-sectional studies clearly described the participants and the research setting. However, the criteria for inclusion were not outlined in three studies, which could lead to confusion for readers regarding the included participants and result in different outcomes. For example, Park, H excluded fragile individuals who were bed-ridden, which probably resulted in an overrated association between social frailty and depressive symptoms [19].

## 4. Discussion

It is noteworthy that the correlation between social frailty and depressive symptoms is documented across both the cohort studies and the cross-sectional studies included in this review. However, studies assessing how these two domains might contribute to each other are scant. This is the first scoping review published to date to show that social frailty is associated with an increased risk of incidence of depressive symptoms. Additionally, the social scales discussed herein are useful for identifying older people at risk of depressive symptoms [10,11].

This association may be partially explained by the fact that many subcomponents of different social frailty scales (going out less frequently than last year (for which “yes” indicates social frailty) [35]; occasionally visiting friends (“no”) and living alone (“yes”) [35,36]; talking with someone everyday (“no”) [37]; constrained financial resources (“a little” or “none at all”); and constrained physical needs (“yes”) [18]) have been linked with depressive symptoms in many previous studies. For example, a previous study which surveyed six lower- and middle-income countries showed that people aged 50 years and above with constrained financial resources and physical needs were more likely to have depressive symptoms due to a lack of support, often due to a decline in familial support and the out-migration of younger adults [18]. Moreover, knowledge from existing studies also shows that socially isolated and lonely older adults often show negative emotions and a high dissatisfaction with life; such older adults do not benefit from communal resources or support for dealing with stressful or adverse life events, contributing to their depression or depressive state [18].

This occurs not only in the general elderly population, but also in elderly individuals with cancer. Previous studies of patients with cancer showed that each subcomponent of social frailty, including living alone [38], social role [39], and social activities [40] such as visiting friends, going out, and talking to someone—is associated with depressive symptoms. Therefore, social frailty may have a strong impact on depressive symptoms in this population. 

This implies that social factors might influence the mental health of the older population. Thus, it is beneficial for researchers to understand the potential effect of existing social intervention programs on the psychological well-being of older adults.

## 5. Limitations

Dependence on study designs. The number of published studies that directly investigated the relationship between social frailty and depression is limited, and the design of included studies are different, which prevented the use of more advanced review methods such as a meta-analysis. Longitudinal data can facilitate the exploration of the potential causal relationship between social frailty and depressive symptoms. However, only 5/14 studies included in our scoping review were cohort studies, showing that social frailty is a risk factor for the incidence of depressive symptoms in older adults. Thus, the results need to be further verified in additional, well-designed cohort studies. Even so, this review presents the current status of the research on this topic and suggests directions for further research.

Issues on the diversity of measurement tools. Across the publications included in this review, a large variety of measurement tools are used to assess social frailty. As a result, the thresholds to define severe social frailty also vary across studies. Thus, the lack of a unified assessment tool and the arbitrary nature of the cutoffs for the social frailty scales might render the results from these studies incomparable. However, this phenomenon does not affect the significant relationship between social frailty and depressive symptoms. These tools are easy for non-healthcare professionals to use in community settings [19], and studies have shown that the social frailty scale may be a more useful predictor for depressive symptoms than the other frailty scales [8]. 

Since the reliability and validity of many scales have yet to be quantified, future research is needed not only to develop a relative unified measure tool of social frailty, but also to test the reliability and validity of the used instruments.

Reliance on the different control variables. Different confounding variables may influence the analysis results. For example, previous studies have shown that diversity in participants (age, sex, education, and marriage status) hampers a more nuanced understanding of the relationship between social frailty and depressive symptoms, and lower socioeconomic status is related to the higher risk of social frailty, which in turn increases the incidence of depressive symptoms [8,10,25,27]. Further research is necessary to elucidate the mechanisms underlying the observed disparate effects of frailty across different control variables. In addition, social frailty can be affected by many environmental factors based on social context. Therefore, research is needed to identify the relationship between the two domains in people of various ethnic groups.

## 6. Conclusions

This scoping review shows that worse social frailty contributes to high depression levels. This indicates that social factors, which are often ignored in preventing the medical context, might represent risk factors for mental health. The timely assessment of and interventions for social frailty may be effective in depressive symptoms.

Our results have potential implications, since social frailty may be prevented or ameliorated. Taking into account the negative impacts of depressive symptoms on both older adults themselves and our society, social frailty should be assessed and community-based interventions taken to relieve the deleterious impact of social frailty on the elderly. Meanwhile, it may be important for medical professionals in hospitals to screen for social frailty in older adults with illnesses, in order to prevent them from developing depressive symptoms.

## Figures and Tables

**Table 1 ijerph-19-16683-t001:** Database search.

Database	Search Term and Strategy	Number of Matches
PubMed	(“social frailty” [Title/Abstract]) AND (depres* [Title/Abstract])	20
Web of Science	“social frailty” (theme) and depres*(theme)	28
CINAHL	AB “social frailty” AND AB depres*	19
EMBASE	(“social frailty” and depres*).mp. [mp = title, abstract, heading word, drug trade name, original title, device manufacturer, drug manufacturer, device trade name, keyword, floating subheading word, candidate term word]	39
Ovid MEDLINE	(“social frailty” and depres*).mp. [mp = ti, ab, tx, ct, sh, ot, nm, hw, fx, kf, ox, px, rx, an, ui, sy]	88
ProQuest	ab (social frailty) AND ab (depress*)	50

## Data Availability

Not applicable.

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
