# Peer review of "The Relationship between Social Frailty and Depressive Symptoms in the Elderly: A Scoping Review"

_ijerph, 2022, doi:10.3390/ijerph192416683_

Round 1
Reviewer 1 Report
It has been a pleasure to review the manuscript “The relationship between social frailty and depressive symptoms in the elders: A scoping review” (manuscript ID ijerph-2015859), submitted to the International Journal of Environmental Research and Public Health.
The research paper addresses a very relevant question, namely the effect of social frailty on depressive symptoms among old-age population. The author carries out a systematic and comprehensive scoping review whose results shows that social frailty has a causal effect on elderly depression. I find that the approach is adequate, and the operationalization of the different concepts is rigorous. I find convincing most of the methodological choices made by the author.
Overall, I find that the paper represents a valuable contribution. My main comments are minor and refer to better contextualize the paper and its relevance. First, I would highlight that the issue can be particularly because of the impact of the pandemic, which has had a significant negative effect on mental health of older adults, even though this segment of population seems to have been more resilient than young groups (Vahia et al., 2020). Second, also in the introduction, I think that the authors should mention that this topic is particularly important given that the current strategies for old-age income security in all types of countries, based on fostering private pensions, do not seem very successful so far in terms of encouraging household savings that warrant decent ageing (Antón et al., 2014; Fadejeva & Tkavecs, 2022). I would suggest that the authors include both arguments and the recommended references in the introductory section.
I believe that, after these minor changes, the paper is suitable for publication.
References
— Antón, J.-I., Muñoz de Bustillo, R.., & Fernández-Macías, E. (2014). Supplementary pensions and saving: evidence from Spain. Journal of Pension Economics and Finance, 13(4),367–388. doi:10.1017/S1474747214000158.
— Fadejeva, L., & Tkavecs, O. (2022). The effectiveness of tax incentives to encourage private savings. Baltic Journal of Economics, 22(2), 110–125. doi: 10.1080/1406099X.2022.2109555
— Vahia, I. V., Jeste, D. V., & Reynolds III, C. F. (2020). Older Adults and the Mental Health Effects of COVID-19. Journal of the American Medical Association, 324(22), 2253–2254. doi:10.1001/jama.2020.21753.
Author Response
Point 1: First, I would highlight that the issue can be particularly because of the impact of the pandemic, which has had a significant negative effect on mental health of older adults, even though this segment of population seems to have been more resilient than young groups (Vahia et al., 2020).
Response 1:
Dear Professor :
Thank you for your advice sincerely. You suggest highlighting that the pandemic has had a significant negative impact on the mental health of older adults. I think this suggestion combines the theme of this study with its relevance perfectly. I have added it in the last sentence of the second paragraph:“This issue has been highlighted by the COVID-19 pandemic, which has had a significant negative effect on the mental health of older adults, even though this segment of population seems to have been more resilient than younger groups ”
Point 2: Second, also in the introduction, I think that the authors should mention that this topic is particularly important given that the current strategies for old-age income security in all types of countries, based on fostering private pensions, do not seem very successful so far in terms of encouraging household savings that warrant decent ageing. (Antón et al., 2014; Fadejeva & Tkavecs, 2022).
Response 2:
Thank you for this suggestion sincerely. I think this advice would make the structure of the background better. I have added it to the last sentence of the background section:“This topic is particularly important given that the current strategies for old-age income security in all types of countries, based on fostering private pensions, do not seem particularly successful thus far in terms of encouraging household savings that warrant decent ageing”
Reviewer 2 Report
abstract
10- restate method, not familiar with the term scoping review
11- change are to were; reword end of sentence to say" 14 studies met the inclusion criteria. 5 should be changed to five
13- the evidence reviewed indicated that 5 studies met category 5, and the others met level 3 criteria
14- disadvantaged social frailty is awkward, reword
16- reword "high depression levels" to - a significant degree of depression
line 22-1. when I read the Background 1st sentence- it sounds like you are focusing on depression, but the abstract states frailty and depression- this needs to be realigned so it sounds more congruent
32- use excluding vs ignoring
33- elderly not old people
34/35- again change old people to elderly
line 36- awkward phrasing- reword
37- emphasized- as an essential component to the health of older adults
38- reword to this: social frailty has been found to lead to various health problems later in life
1-39- spacing around parentheses doesn't seem correct
There continues to be grammatical corrections needed; from this point forward, will identify those sentences needing rewording, an English editor may be helpful
line 41- clarify the multiple adjustments- sounds awkward and a little vague
line44- change ,that is, to ;that is,
line 51- reword to -which has produced
lines 54 and 55- reword this
line 59- has a greater impact for evidence-based practice in nursing
lines 62-63- awkward wording-reword
line 62-63- liked the explanation- just reword
line 65- don't say to the best of your knowledge, just explain why you think that i.e.- a thorough and exhaustive review of the literature was completed and ....
line 74- not is but was conducted
line70- omit thorough, it is assumed that is what you did
like table 1!!
line 82- change are to were
line 84-change are to were
from this point on, I will not correct tense, but should be past tense in general as you have completed this study
line 105- refers to table 2 and 3 which are not included
like paragraph 3.1- nice explanation
line 124- so if 4 studies did not define social frailty, why did you include them- explain why- I am assuming that the studies were of appropriate rigor in spite of this omission
lines 163-4- while the cut offs are slightly different between studies, they are largely very similar
lines 191-2- reword the last sentence, I would think you are looking for social frailty and depression, explain that statement
line 256- again, don't say to the best of our knowledge, you have done a rigorous search, so just say this is the first scoping review published to date...
line 270- in what population
general comment- a little more discussion I think is in order for all the work you did on this
limitations- do like this section and appropriate based on the type of review done- great !!
line 324- not sure what upstream is, but don't you mean that social frailty is a risk factor for other comorbidities, and understanding how to impact this condition is important.
great topic, just needs rewording in multiple sections
line 312- make further research a separate paragraph
Author Response
Dear Professor:
Thank you sincerely for your advice. I admire your patience and carefulness inthis study. Thank you again for your hard work in this research ! I will respond to your valuable suggestions one by one.
Point 1:10- restate method, not familiar with the term scoping review
Response 1: I have repeated the scoring review in more detail.
Point 2:11- change are to were; reword end of sentence to say" 14 studies met the inclusion criteria. 5 should be changed to five
Response 2: I have revised.
Point 3:13- the evidence reviewed indicated that 5 studies met category 5, and the others met level 3 criteria
Response 3: I have revised.
Point :14- disadvantaged social frailty is awkward, reword
Response : reworded “This scoping review shows that worse social frailty contribute to ...”
Point :16- reword "high depression levels" to - a significant degree of depression
Response : reworded
Point :line 22-1. when I read the Background 1st sentence- it sounds like you are focusing on depression, but the abstract states frailty and depression- this needs to be realigned so it sounds more congruent
Response : I have realigned in the first sentence of the background and add a sentence“Depressive symptoms and social frailty are increasingly popular in geriatric research. The relationship between the two topics is of great interest to researchers. ”
Point :32- use excluding vs ignoring
Response : revised.
Point :33- elderly not old people
Response : revised.
Point :34/35- again change old people to elderly
Response :revised.
Point :line 36- awkward phrasing- reword
Response :reworded
Point :37- emphasized- as an essential component to the health of older adults
Response : have emphasized. “Recently, a systematic review showed that not only physical and cognitive aspects, but also social aspects of frailty are essential components of health in the elderly.”
Point :38- reword to this: social frailty has been found to lead to various health problems later in life
Response : reworded. “In fact, social frailty has been found to lead to various health problems later in life”
Point :1-39- spacing around parentheses doesn't seem correct
There continues to be grammatical corrections needed; from this point forward, will identify those sentences needing rewording, an English editor may be helpful
Response : I have revised the grammatical corrections, and ask an English editor in your editing services for help.
Point :line 41- clarify the multiple adjustments- sounds awkward and a little vague
Response : clarified. “after adjusting for sociodemographic factors(e.g., age, gender, smoking, alcohol intake, etc. )”
Point :line44- change ,that is, to ;that is,
Response :changed
Point :line 51- reword to -which has produced
Response : reworded.
Point :lines 54 and 55- reword this
Response : reworded.
Point :line 59- has a greater impact for evidence-based practice in nursing
Response : revised
Point :lines 62-63- awkward wording-reword
Response : reworded.
Point :line 62-63- liked the explanation- just reword
Response : reworded.
Point :line 65- don't say to the best of your knowledge, just explain why you think that i.e.- a thorough and exhaustive review of the literature was completed and ....
Response : reworded.“A thorough and exhaustive review of the literature was completed and there is no review addressing our research question.”
Point :line 74- not is but was conducted
Response :reworded.
Point :line70- omit thorough, it is assumed that is what you did like table 1!!
Response : omitted.
Point :line 82- change are to were
Response : changed
Point :line 84-change are to were from this point on, I will not correct tense, but should be past tense in general as you have completed this study
Response : changed
Point :line 105- refers to table 2 and 3 which are not included
Response :Tables 2 and 3 are in the appendix, and I have changed Tables 2 and 3 to Tables A1 and A2 according to the IJERPH template.
Point :like paragraph 3.1- nice explanation
Response : Thank you very much for your approval
Point :line 124- so if 4 studies did not define social frailty, why did you include them- explain why- I am assuming that the studies were of appropriate rigor in spite of this omission
Response : have explained as“However, another four studies did not provide a definition for social frailty, though they were of appropriate rigor in spite of this omission. ”
Point :lines 163-4- while the cut offs are slightly different between studies, they are largely very similar
Response : revised “While the cut offs were slightly different between studies, they were largely very similar.”
Point :lines 191-2- reword the last sentence, I would think you are looking for social frailty and depression, explain that statement
Response : explained as “However, there were positive relationships between social frailty and depressive symptoms in all of the findings, even though the shortest follow-up time was only 1 year.”
Point :line 256- again, don't say to the best of our knowledge, you have done a rigorous search, so just say this is the first scoping review published to date...
Response : revised
Point :line 270- in what population
general comment- a little more discussion I think is in order for all the work you did on this.
limitations- do like this section and appropriate based on the type of review done- great !!
Response : have discussed more in the population, and thank you very much for your approval for the limitations.
Point :line 324- not sure what upstream is, but don't you mean that social frailty is a risk factor for other comorbidities, and understanding how to impact this condition is important.
Response : revised as “social frailty should be assessed and take the community-based interventions to relieve the deleterious impact of social frailty in the elderly.”
Point :line 312- make further research a separate paragraph
Response : have adjusted further research in a separate paragraph